# Modeling Label Correlatons Implicitly Through Latent Label Encodings for Multi-label Text Classification

## Abstract

Multi-label text classification (MLTC) aims to assign a set of labels to each given document. Unlike single-label text classification methods that often focus on document representation learning, MLTC faces a key challenge of modeling label correlations due to complex label dependencies. Previous state-of-the-art works model label correlations explicitly. It lacks flexibility and is prone to introduce inductive bias that may not always hold, such as label-correlation simplification, sequencing label sets, and label-correlation overload. To address this issue, this paper uses latent label representations to model label correlations implicitly. Specifically, the proposed method concatenates a set of latent labels (instead of actual labels) to the text tokens, inputs them to BERT, then maps the contextual encodings of these latent labels to actual labels cooperatively. The correlations between labels, and between labels and the text are modeled indirectly through these latent-label encodings and their correlations. Such latent and distributed correlation modeling can impose less a priori limits and provide more flexibility. The method is conceptually simple but quite effective. It improves the state-of-the-art results on two widely used benchmark datasets by a large margin. Further experiments demonstrate that its effectiveness lies in label-correlation utilization rather than document representation. Feature study reveals the importance of using latent label embeddings. It also reveals that contrary to the other token embeddings, the embeddings of these latent labels are sensitive to tasks; sometimes pretraining them can lead to significant performance loss rather than promotion. This result suggests that they are more related to task information (i.e., the actual labels) than the other tokens. [1]

## 1 Introduction

Multi-label text classification (MLTC) aims to associate each text sample with a set of labels. It can be applied to a wide spectrum of scenarios, including text categorization (Schapire & Singer, 2000), information retrieval (Gopal & Yang, 2010) and web mining (Zhang & Zhou, 2013; Liu et al., 2020). Unlike single-label text classification, in MLTC a text is usually associated with more than one label. Typically, these labels correlate with each other. Utilizing their correlations effectively can benefit their predictions significantly (Tsoumakas et al., 2009; Gibaja & Ventura, 2015). However, due to the high-order nature of these correlations, it is a challenging task (Zhang & Zhou, 2013).

There are different ways to deal with this challenge. A relatively earlier line of works models the correlations independently of text context (Prabhu & Varma, 2014; Jernite et al., 2017; Peng et al., 2018; Singh et al., 2018). A representative class in this line is the tree-structure methods. Typically, they build a hierarchical structure for the label space statistically or according to taxonomy. During predictions, they usually utilize the modeled label correlations through some regularization methods (Peng et al., 2018; Singh et al., 2018). Since the structure of labels that heavily limits the correlations is defined in advance, they have introduced a strong inductive bias that suggests that labels correlate with each other following a tree structure unrelated to text context. But in real tasks, the labels and the text often correlate with each other closely in a complicated way. The correlations between

---

[1]The source code is provided in the supplementary file.

labels can be quite different depending on text contexts. It may not be appropriate to simplify their correlations a priori too much.

Another representative class of works in this line is the label embedding methods. In computer vision, researchers have used label embeddings to capture label correlations to enhance performances on zero-shot learning and other tasks (Palatucci et al., 2009; Yogatama et al., 2015; Ma et al., 2016). Inspired by them, Wang et al. (2018) introduce joint label embeddings to model label relations and label-text relations to enhance text classification performances. Their work is extended by Xiao et al. (2019). In these works, the correlations between labels are modeled by the similarities between label embeddings; they are learned adaptively. Such correlation modeling is more flexible and expressive compared with the tree structure methods. And they do not only model label correlations but also model label-text relations. However, they still model label correlations independently of context.

A relatively recent line of works not only embeds labels but also encodes them contextually through attention-based encoding (or encoding-decoding) mechanisms. Such mechanisms allow them to jointly model the correlations between labels, and between labels and the text context with less a priori limits. This line of works can be further divided into three major classes: sequence-to-sequence (Nam et al., 2017; Yang et al., 2018), sequence-to-set (Yang et al., 2019; Tsai & Lee, 2020) and joint encoding (Zhang et al., 2021).

The sequence-to-sequence methods are first in the line. They are based on the well-known sequence-to-sequence framework. Generally, they encode text by deep neural networks and then decode them to targeted labels one by one. They lead to a breakthrough in this area by eliminating the inductive bias that label correlations are independent of text context. But due to the nature of the classic seq2seq framework, they also introduce another inductive bias: there is an order among the labels associated with each text. However, it is difficult to define such orders appropriately. And this issue is critical to these methods (Vinyals et al., 2015).

To overcome the difficulties in defining appropriate label orders, the sequence-to-set methods (Yang et al., 2019; Tsai & Lee, 2020) are proposed. Different from seq2seq methods that calculate loss each time a label is generated, they use reinforcement learning to delay loss calculation to the time when the whole label sequence is generated. Since the label sequences are constructed adaptively during training by reinforcement learning, it can obtain them more reasonably. However, the labels are essentially a set rather than a sequence. Constructing the label sequence adaptively still regards the labels as a sequence rather than a set. It does not eliminate the label-sequence inductive bias completely.

Very recently, a new method (Zhang et al., 2021) (it is the most related work to ours) encodes the text and the labels jointly within a unified encoding process using BERT (Devlin et al., 2018). It performs extra tasks on the label encodings to further model label correlations. By doing so, it removes the label-sequence inductive bias and obtains state-of-the-art results. But their work still suffers from the following three issues. (1) Their extra tasks are designed based on the following consideration: in the context of a given text, the co-occurred labels should have stronger mutual correlations. Since unrelated but common labels can co-occur frequently, we think this inductive bias may still be excessive. We call this issue label-correlation overload. (2) Their extra task *plcp* models label-correlations pairwisely rather than globally; it may also be an oversimplification. (3) Although jointly encoding actual labels and texts has addressed the label-sequence inductive bias elegantly, it can be less effective and even impossible when the number of actual labels is very large. This issue limits its applications heavily.

All the above methods share a common fundamental characteristic: they model label correlations explicitly. As one may observe from the above discussions, due to the high-order, contextual and non-sequential nature of label correlations, it is difficult to model them explicitly without introducing improper inductive bias. To overcome this difficulty, this paper uses latent label encodings to model actual labels and their correlations implicitly. Specifically, it pads a set of latent labels rather than actual labels in front of the inputs to BERT, and then maps the contextual encodings of these latent labels to actual class labels cooperatively. The label-label and label-text correlations regarding the actual labels are modeled indirectly through these latent-label encodings and their correlations. Since the correlations between these latent-label encodings, and their relations with actual labels and the text context are all trained adaptively, it imposes fewer a priori limits and provides more flexibility.

Furthermore, by encoding latent labels rather than actual labels, our method can well address the forementioned three issues of the most related work to ours (Zhang et al., 2021): (1) Our label-correlations are learned adaptively with few a priori limits, this can avoid label-correlation overload introduced by empirical extra tasks; (2) By extensively utilizing latent-label encodings and their relations in each label's prediction, we address the oversimplification of pairwise label-correlation modeling. (3) The number of latent-label is an adjustable hyper-parameter that can be adjusted to accommodate specific tasks, usually it is much smaller than the number of actual labels. This feature addresses the issue of massive-label well.

General-purpose text classification algorithms are not specially designed for MLTC, such as TextCNN (Kim, 2014) and BERT (Devlin et al., 2018). They do not consider label correlations. In a sense, they can also be regarded as modeling label correlations implicitly. We think that the advantage of our algorithm over them in capturing label correlations lies in the following aspects: compared with their latent neurons, our latent label encodings are structural and high-orderly correlated. These features make it more expressive in modeling label correlations, just like the actual label encodings do.

We summarize the major contributions of this paper as follows:

(1) To the best of our knowledge, all the previous state-of-the-art works for MLTC regard label correlations as a key challenge. They model label correlations explicitly, which lacks flexibility and hence is prone to suffer from inappropriate inductive bias. Differently, we use latent label encodings to model label correlations implicitly. The difference between our work and the previous state-of-the-art works in modeling label correlations is fundamental, it provides a new line of thought.

(2) Our algorithm outperforms the state-of-the-art results on two widely used benchmark datasets by a large margin. Feature study shows that it significantly outperforms using actual label encodings for classification. Error analysis indicates that the advantage of our algorithm lies in label-correlation utilization rather than document representation. The inspiring results demonstrate the great potential of modeling label correlations implicitly.

(3) Feature study further reveals that contrary to the other token embeddings, sometimes pretraining latent label embeddings can lead to significant performance loss instead of promotion. This suggests that they are more related to task information (i.e., the actual labels) than the other tokens.

## 2 METHODOLOGY

Given a text $T$ containing $m$ sequential word tokens $w_0, w_1, \ldots, w_{m-1}, w_i \in \mathbb{V}(0 \leq i \leq m-1)$, a MLTC task aims to determine a label set $\mathbb{LS} = \{l_0, l_1, \ldots, l_{p-1}\}, l_j \in \mathbb{L}(0 \leq j \leq p-1, |\mathbb{L}| = n)$, $m$ is the length of the text, $\mathbb{LS}$ is the set of labels associated with the text, $p$ is the number of labels in $\mathbb{LS}$, $\mathbb{V}$ is the vocabulary of word tokens, $\mathbb{L}$ is the complete set of all the candidate labels, $n$ is the cardinality of $\mathbb{L}$. Our method to solve this problem is straightforward: it pads $k$ latent labels $\boldsymbol{ll} = (ll_0, ll_1, \ldots, ll_{k-1})$ in front of the input to BERT and maps the encodings of these latent labels to actual labels through a dense neural network. Figure 1 shows its framework. In the following, we describe them in detail.

**Input**

As illustrated in Figure 1, the input is a concatenation of latent labels and text tokens. Specifically, let the latent labels be $\boldsymbol{ll} = (ll_0, ll_1, \ldots, ll_{k-1})$, the tokenized text be $(w_0, w_1, \ldots, w_{m-1})$, the input then is $(ll_0, ll_1, \ldots, ll_{k-1}, w_0, w_1, \ldots, w_{m-1}, [\text{SEP}])$. These latent labels do not exist in the vocabulary of BERT; their initial embeddings are generated randomly. The number $k$ of latent labels is a hyperparameter. Since all these latent labels share the same position embedding 0, they are a set rather than a sequence. Usually, the number of latent labels is much smaller than that of the actual labels. It makes the input to BERT more concise and easier to train compared with using actual label embeddings. Also, the condensed representations of latent labels makes it easier for them to correlate with each other closely.

**BERT within-task further pretraining (optional)**

The setting of the BERT part is the same as traditional BERT. It is well known that in-domain further pretraining BERT can enhance its adaptability to the specific contexts. So, we employ an optional

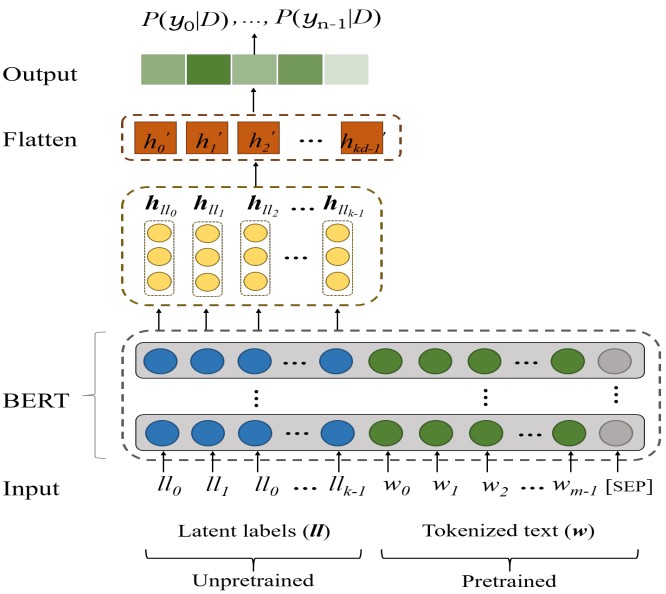

Figure 1: The framework of LLEM.

within-task further pretraining process that follows the within-task pretraining procedure introduced in (Sun et al., 2019). Since the latent labels are task-targeted, their embeddings are not pretrained during this process as other tokens do. As shown by our experiment, pretraining them can decrease performance significantly (see 4.2). Note that this is only an optional process; our algorithm can still outperform the state-of-the-art results by a large margin without this process.

**Output and Loss**

As illustrated in Figure 1, let the contextual encodings of these latent labels be $\boldsymbol{H}_l = [\boldsymbol{h}_{ll_0}, \boldsymbol{h}_{ll_1}, \ldots, \boldsymbol{h}_{ll_{k-1}}]$. They are first flattened and then fed to a single layer dense network to obtain the predictions:

$$\hat{\boldsymbol{y}} = sigmoid(\boldsymbol{W}_o^T flatten(\boldsymbol{H}_l) + \boldsymbol{b}_o) \tag{1}$$

$\boldsymbol{W}_o \in \mathbb{R}^{kd \times n}$ and $\boldsymbol{b}_o \in \mathbb{R}^n$ are the trainable weights and bias of the dense neural network respectively, $d$ is the dimension of the encodings, $n$ is the number of actual labels. For the recent state-of-the-art LACO algorithm (Zhang et al., 2021), its final prediction on each label is determined only by the encoding of the label itself. On the contrary, our prediction on each label is determined by all the encodings of these latent labels cooperatively. The advantage of our method is twofold. Firstly, it utilizes not only the label encodings but also their correlations. Secondly, the encodings of these latent labels are shared among all the predictions; compared with predicting each label separately, this general sharing can enhance robustness.

Like previous works, we use Binary Cross Entropy as the loss function for the multi-label text classification tasks:

$$\mathcal{L}_{mltc} = -\sum_{i=1}^{n} [q_i \ln p_i + (1 - q_i) \ln (1 - p_i)] \tag{2}$$

where $p_i = P(y_i|D)$ is the probability of label $i$ predicted by the model $(y_i \in \mathbb{L})$, and $q_i \in \{0, 1\}$ indicates whether $y_i \in \mathbb{LS}$.

# 3 EXPERIMENT

## 3.1 DATASETS

We evaluate our Latent Label Encoding Method (LLEM) on two widely used multi-label text classification datasets: AAPD (Yang et al., 2018) and RCV1-V2 (Lewis et al., 2004). The AAPD (Arxiv

Academic Paper) dataset includes 55,840 abstracts of papers involving 54 topics in computer science. RCV1-V2 (Reuters Corpus Volume I) consists of 804,414 manually categorized newswire stories involving 103 topics. We followed the same data-splitting as Yang et al. (2018).

## 3.2 Experiment Setting

We implement our model using Pytorch and run it on 1 NVIDIA Tesla V100 (32G). The model is based on the English base-uncased versions of BERT, just as other BERT-based MLTC models do (Cai et al., 2020; Zhang et al., 2021). The BERT-base model can be further within-task pretrained on the training set to enhance its adaptability to specific contexts (optional) (Sun et al., 2019).

The hyper-parameters of the optional within-task pretraining of the BERT-based model are set according to Sun et al. (2019). Specifically, the batch size is 32, the max sequence-length is 128, the warm-up steps are 10,000, the learning rate is 5e-5, and the train steps are 100,000.

The training hyper-parameters of our method are tuned by monitoring Micro-F1 score on the validation set. They are set as follows. The number of latent label is 15 and 20 for AAPD and RCV1-V2 respectively, the batch size is 24, the dropout probability is 0.1, and the max sequence-length is 512. We use Adam method, and set $\beta_1 = 0.9, \beta_2 = 0.999$ respectively. For sequences longer than 512 (including latent labels), we use head+tail truncation proposed by Sun et al. (2019) with trunc medium = 128. Following Howard & Ruder (2018) and Sun et al. (2019), we utilize layer-learning rate decay strategy with decay factor = 0.95. We also use slanted triangular (Howard & Ruder, 2018) with base learning rate set as 5e-5, and the warm-up proportion as 0.1. The total training epochs are set as 100 and 8 for AAPD and RCV1-V2 respectively. We save the best model on the validation set for testing.

## 3.3 Evaluation Metrics

For MLTC tasks, a text is associated with more than one label. The objective is not to find such "a" label correctly, but to find them "all" correctly. Therefore, the recall metric is also important in addition to precision. So instead of precision , the comprehensive metrics Micro-F1 and Hamming Loss are the most important metrics for MLTC tasks (Chen et al., 2017; Yang et al., 2018). Micro-F1 is the weighted average of F1 scores of each class label, while Hamming Loss represents the fraction of misclassified instance-label pairs. They are good metrics to evaluate the overall performance of an MLTC algorithm. Macro-F1 is more sensitive to performance on low-frequency labels. Due to the deficiency of data, the performance on low-frequency labels is highly dependent on effective usage of label correlations. Therefore, Macro-F1 is a good tool to analyze the capacity of an algorithm in utilizing label correlations. Besides these metrics, Micro/Macro-Precision and Micro/Macro-Recall are also reported for reference.

## 3.4 Comparing Algorithms

We compare our algorithm with a wide spectrum of competitive algorithms, including the well-known general-purpose text-classification algorithms: Binary Relevance (BR) (Boutell et al., 2004), TextCNN (Kim, 2014) and BERT (Devlin et al., 2018); label embedding algorithms: LEAM (Wang et al., 2018) and LSAN (Xiao et al., 2019); the sequence generation model (SGM) (Yang et al., 2018); the deep reinforced sequence-to-set model (Yang et al., 2019); the OCD framework (Tsai & Lee, 2020); the ML-Reasoner (Wang et al., 2021); and two recent state-of-the-art BERT based algorithms: the hybrid-BERT (Cai et al., 2020) and LACO (+*plcp, +clcp*) (Zhang et al., 2021). To the best of our knowledge, these algorithms have provided all the previous state-of-the-art results on the two datasets regarding major metrics.

Among these algorithms, LACO (+*plcp, +clcp*) is most related to our Latent Label Encoding Method (LLEM). Both our works encode texts and labels jointly and are based on the BERT-base pretraining model. The key differences are as follows: (1) LACO encodes the actual labels in the same space of text tokens to capture label-label and label-text correlations. On the contrary, our method encodes the latent labels rather than the actual labels, the correlations regarding actual labels are modeled indirectly through these latent-label encodings and their relations. (2) In their work, the encodings of labels are used for prediction (and extra tasks) separately; differently, in our work correlations between the latent label encodings are utilized globally and shared by all the

Table 1: Performance of different algorithms on the two datasets. Evaluation metrics include: Hamming Loss (HL), Micro (Mi-) and Marco (Ma-) average Precision (P), Recall (R), F1-Score (F1). Symbol ↑ denotes higher is better, and vice versa.

| Algorithm | AAPD dataset | | | RCV1-V2 dataset | | |
|---|---|---|---|---|---|---|
| | HL ↓ | Mi- P / R / F1 ↑ | Ma- P / R / F1 ↑ | HL ↓ | Mi- P / R / F1 ↑ | Ma- P / R / F1 ↑ |
| BR(Boutell et al., 2004) | 0.0316 | 64.4 / 64.8 / 64.6 | - - - | 0.0086 | 90.4 / 81.6 / 85.8 | - - - |
| TextCNN(Kim, 2014) | 0.0256 | **84.9** / 54.5 / 66.4 | - - - | 0.0089 | 92.2 / 79.8 / 85.5 | - - - |
| BERT(Devlin et al., 2018) | 0.0224 | 78.6 / 68.7 / 73.4 | 68.7 / 52.1 / 57.2 | 0.0073 | 92.7 / 83.2 / 87.7 | 77.3 / 61.9 / 66.7 |
| LEAM(Wang et al., 2018) | 0.0261 | 76.5 / 59.6 / 67.0 | 52.4 / 40.3 / 45.6 | 0.0090 | 87.1 / 84.1 / 85.6 | 69.5 / 65.8 / 67.6 |
| LSAN(Xiao et al., 2019) | 0.0242 | 77.7 / 64.6 / 70.6 | 67.6 / 47.2 / 53.5 | 0.0075 | 91.3 / 84.1 / 87.5 | 74.9 / 65.0 / 68.4 |
| SGM(Yang et al., 2018) | 0.0251 | 74.6 / 65.9 / 69.9 | - - - | 0.0081 | 88.7 / 85.0 / 86.9 | - - - |
| Seq2Set(Yang et al., 2019) | 0.0247 | 73.9 / 67.4 / 70.5 | - - - | 0.0073 | 90.0 / 85.8 / 87.9 | - - - |
| OCD(Tsai & Lee, 2020) | - | - - 72.0 | - - 58.5 | - | - - - | - - - |
| ML-R(Wang et al., 2021) | 0.0248 | 72.6 / **71.8** / 72.2 | - - - | 0.0079 | 89.0 / 85.2 / 87.1 | - - - |
| HBLA(Cai et al., 2020) | 0.0223 | 76.8 / 72.2 / 74.4 | - - - | 0.0063 | 90.6 / 89.2 / 89.9 | - - - |
| LACO(Zhang et al., 2021) | 0.0213 | 80.2 / 69.6 / 74.5 | 70.4 / 54.0 / 59.1 | 0.0072 | 90.8 / 85.6 / 88.1 | 75.9 / 66.6 / 69.2 |
| LACO+*plcp*(Zhang et al., 2021) | 0.0212 | 79.5 / 70.8 / 74.9 | 68.4 / 55.8 / 59.9 | 0.0070 | 90.8 / 86.2 / 88.4 | 76.1 / 66.5 / 69.2 |
| LACO+*clcp*(Zhang et al., 2021) | 0.0215 | 78.9 / 70.8 / 74.7 | **71.9** / 56.6 / 61.2 | 0.0070 | 90.6 / 86.4 / 88.5 | 77.6 / 71.5 / 73.1 |
| LLEM | 0.0207 | 80.1 / 71.6 / 75.6 | 68.3 / 57.6 / 61.5 | **0.0056** | 92.8 / **89.3** / **91.0** | 78.5 / **75.3** / **76.0** |
| LLEM+*wpre* | **0.0203** | 80.9 / 71.7 / **76.0** | 71.8 / **59.0** / **63.0** | **0.0056** | **93.7** / 88.2 / 90.8 | **79.2** / 71.2 / 73.8 |

predictions; (3) Our work has not performed any extra tasks to further capture label correlations as Zhang et al. (2021); instead, our work has performed an optional unsupervised within-task further pretraining.

# 4 RESULTS AND ANALYSIS

In this section, we provide the experimental results, and then perform an in-depth analysis on them and our method.

## 4.1 EXPERIMENTAL RESULTS

We list the experimental results of our algorithms and the compared baselines on the two datasets in Table 1. To the best of our knowledge, the results of the compared baselines include all the previous state-of-the-art results on the two datasets regarding major metrics. Our algorithms for comparison are as follows. (1) LLEM; (2) LLEM+*wpre*: LLEM that have all the token embeddings (except for the latent label embeddings) within-task pretrained.

From the results, one can observe that our algorithms are quite competitive. It outperforms the previous state-of-the-art results on both datasets by a large margin regarding the two major metrics Micro-F1 and Hamming Loss. Specifically, for the AAPD dataset, LLEM+*wpre* outperforms the previous best Micro-F1 (provided by LACO+*plcp*) by $1.47\%$, previous best Hamming Loss (also provided by LACO+*plcp*) by $4.25\%$; for the RCV1-V2 dataset, LLEM+*wpre* outperforms the previous best Micro-F1 (provided by hybrid BERT) by $1.00\%$, previous best Hamming Loss (also provided by hybrid BERT) by $11.11\%$. Even without within-task pretraining, our base LLEM still outperforms the corresponding state-of-the-art results on AAPD by $0.93\%$ and $2.42\%$, the corresponding state-of-the-art results on RCV1-V2 by $1.22\%$ and $11.11\%$, respectively. This indicates that our method can achieve substantial improvements by modeling label correlations implicitly through latent label encodings.

## 4.2 FEATURE STUDY

The objective of the feature study is to evaluate whether using latent labels is value-added compared with using actual labels, as well as investigate the effect of pretraining on latent label embeddings. To this end, we compare our algorithm LLEM+*wpre* with two baselines. The first baseline using actual label embeddings instead of latent label embeddings and predict each label based on its encoding; we name it as Actual Label Encoding Model (ALEM+*wpre*). The second baseline is LLEM+*wpre_all*, i.e., LLEM that have all the token (including the latent label) embeddings within-task pretrained. Besides the mentioned difference, all the settings for these compared algorithms are the same. We

Table 2: Feature study results. Comparing LLEM+*wpre* with two baselines ALEM+*wpre* and LLEM+*wpre_all* to study the value of latent label encoding, and the effect of pretraining on latent label embeddings.

| | AAPD | | | RCV1-V2 | | |
|---|---|---|---|---|---|---|
| **Model** | **HL** ↓ | **Mi-F1** ↑ | **Ma-F1** ↑ | **HL** ↓ | **Mi-F1** ↑ | **Ma-F1** ↑ |
| LLEM+*wpre* | 0.0203 | 76.01 | 63.05 | 0.0056 | 90.84 | 73.84 |
| ALEM+*wpre* | 0.0227 | 73.84 | 59.53 | 0.0058 | 90.51 | 71.93 |
| LLEM+*wpre_all* | 0.0214 | 74.83 | 59.48 | 0.0057 | 90.59 | 73.21 |

compare these algorithms on the two datasets based on the major metrics, i.e., Hamming Loss, Micro-F1 and Macro-F1. The results are listed in Table 2.

The results show that our algorithm significantly outperforms ALEM+*wpre*. It indicates that it is value-added by modeling label correlations implicitly. Compared with Micro-F1, the gap between the two algorithms is even bigger on the Macro-F1 metric, this means the advantage of latent label encoding is more prominent for less frequent labels, sugesting that it is more superior in label-correlation utilization. The advantage of our algorithm is also more notable on the AAPD dataset. We think capturing label correlations is more difficult for this dataset due to its much smaller size and more professional text type. This makes the capacity of our method in label-correlation utilization more valuable and hence can lead to more prominent results.

One can also observe that LLEM+*wpre* outperforms LLEM+*wpre_all* significantly on the more challenging dataset AAPD. We think it is also attributed to the difficulties in capturing label-correlations in AAPD. In such difficult cases, the latent labels should relate to actual labels more closely to better capture their correlations and contribute to their predictions. But pretraining the latent label embeddings without knowing the actual labels cannot contribute to this purpose at all; it can even misguide the latent-label embeddings and make the connection between latent labels and actual labels more difficult. This result suggests that compared with the other token embeddings, the latent label embeddings are sensitive to specific tasks; they are more related to task information (i.e., the actual labels) rather than texts in certain cases.

## 4.3 ERROR ANALYSIS

LACO (and its extensions LACO+*plcp*, LACO+*clcp*) is the latest state-of-the-art baseline. It is the most related work to our method. LACO+*plcp* also has provided the previous best results on AAPD, which is more challenging in label-correlation utilization due to its much smaller size and more professional text type. In this subsection, we further perform two error analyses on AAPD to provide more in-depth comparisons between LLEM+*wpre* and LACO+*plcp* on their label-correlation utilization. We use Macro-F1 as the comparing metric since it is more sensitive to label-correlation utilization, as discussed in 3.3. These two error-analyses compare the two algorithms on labels with different frequencies (F), as well as on samples with different number of labels (L). The design intuition of these two analyses are as follows. Due to the deficiency of data, the performance on low frequency labels are always much lower, their performances are highly dependent on effective usage of label correlations (Yang et al., 2019; Menon et al., 2020). And for intensive-label samples, it is difficult to predict all their labels correctly and completely. But the co-existing labels can contribute to each other's predictions through their correlations. Therefore, an algorithm that can make effective usage of label-correlations should be capable of providing good results on low-frequency labels, as well as providing good results on intensive-label samples.

The first error analysis compares our LLEM+*wpre* with LACO+*plcp* on labels with different frequencies. We divide the labels into four groups according to their frequencies, just as Zhang et al. (2021): the first group: F > 4500; the second group: $1700 < F \le 4500$; the third group: $870 < F \le 1700$; and the last group: $F \le 870$. The results are listed in Fig 2(a). As the frequency decreases, the performances of the two algorithms both decrease notably, indicating the difficulty in predicting low-frequency labels. The two algorithms provide similar results on higher-frequency labels (F > 1700), indicating that they have similar capacities in document representation. But for the lower-frequency labels ($F \le 1700$), our algorithm provides notably higher performance, suggest-

ing that it is superior in label-correlation utilization. Taken together, these results suggest that the advantage of our algorithm lies in label correlation utilization rather than document representation.

The second error analysis compares LLEM+*wpre* with LACO+*plcp* on samples with different numbers of labels. We divide the samples into three groups according to their number of labels (L): the first group: L ≤ 2; the second group: L = 3; and the last group: L ≥ 4. The results are listed in Fig 2(b). As the number of labels (L) increases, both performances decrease notably, indicating that it is difficult to handle intensive-label samples. The two algorithms provide close results on the sparsest-label samples (L ≤ 2). For such sparsest-label samples, it is very close to single-label text classification. This further indicates that the two algorithms have similar capacities in document representation. But for the more intensive-label samples (L ≥ 3), our algorithm again shows significantly higher performance, indicating that it is more effective in label-correlation utilization. To sum up, these results further demonstrate that the advantage of our algorithm lies in label correlation utilization rather than document representation.

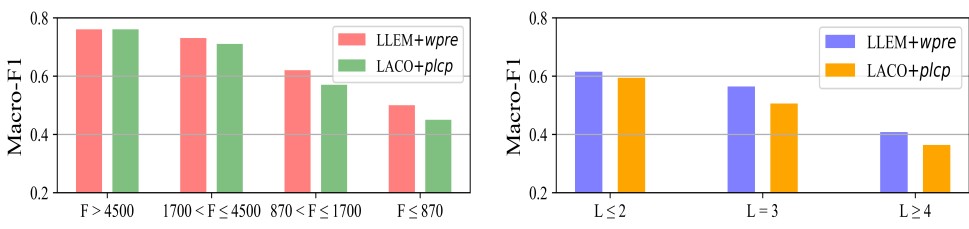

(a) Error analysis on labels with different frequencies (F)

(b) Error analysis on samples with different number of labels (L)

Figure 2: Comparative error analysis of our method (LLEM+*wpre*) and the state-of-the-art (LACO+*plcp*) on the more challenging dataset AAPD.

## 5 RELATED WORKS

The research on multi-label text classification follows two directions: enhancing document representation and modeling label correlations.

**Document representation.** Document representation is common for all NLP tasks. Basically, the developments in document representation for MLTC are inspired by other areas of NLP. These include RNN based models (Liu et al., 2016), CNN based models (Kurata et al., 2016; Liu et al., 2017), combining CNN and RNN in models (Chen et al., 2017; Lai et al., 2015), attention-mechanism based models (Yang et al., 2016; You et al., 2019; Adhikari et al., 2019), and most recently the powerful pretrained BERT (Devlin et al., 2018) based methods (Cai et al., 2020; Zhang et al., 2021).

**Modeling label-correlations.** Different from single-label text classification (SLTC), multi-label text classification (MLTC) faces a key challenge of modeling label correlations. There are two major lines of works to this end. The first line of works model label-correlations separately to document representation. These include the methods transforming multi-label text classification to multi or binary class text classifications (Tsoumakas & Katakis, 2007; Read et al., 2011), the tree-structure methods (Prabhu & Varma, 2014; Jernite et al., 2017; Peng et al., 2018; Singh et al., 2018), the label embedding methods (Wang et al., 2018; Xiao et al., 2019), and the correlation neural network method (Xun et al., 2020). Since labels and the text often correlate with each other complicatedly, it can be over-simplified to separate these two processes. The other line of works integrates label-correlation modeling with document representation by encoding labels contextually with attention mechanisms. These methods include sequence-to-sequence (Nam et al., 2017; Yang et al., 2018), sequence-to-set (Yang et al., 2019; Tsai & Lee, 2020), and most recently the joint encoding with multi-tasks (Zhang et al., 2021). Although they have overcame previous over-simplification inductive bias, they suffer from issues such as label-sequence and label-correlation overload.

## 6 CONCLUSIONS

How to model label correlations effectively is a specific and important challenge to multi-label text classification. Previous state-of-the-art works for MLTC model label correlations explicitly. It lacks flexibility and hence is prone to introduce inductive bias that may not always hold. Differently, this paper proposes a method to model label correlations implicitly through latent label encodings. The difference between our work and previous state-of-the-art works in modeling label-correlations is fundamental; it provides a new line of thought. The method outperforms the state-of-the-art results on two widely used benchmarks by a large margin. Analyses also show that the effectiveness of our method lies in label-correlation utilization rather than document representation learning. The inspiring results demonstrate the great potential of modeling label correlations implicitly.

The other contributions of this paper include but are not limited to: the concepts of sparse-label (SL) samples and intensive-label (IL) samples for analyzing label-correlation utilization; revealing the singular nature of latent label embeddings compared with the other token embeddings regarding pretraining.

In future works, we think it is intresting for us and the other researchers to investigate how the latent-label correlate with each other, with the actual labels and the context, and how can they be interpreted. They certainly can shed light on new methods of their design and training.

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
