# OpenReview forum: "Modeling label correlations implicitly through latent label encodings for multi-label text classification"
_ICLR.cc/2022/Conference — ICLR 2022 Submitted_

### Official Review · Reviewer_8DdA · 2021-10-30

**Correctness:** 3
**Technical Novelty And Significance:** 3
**Empirical Novelty And Significance:** 3
**Recommendation:** 6
**Confidence:** 4

**Main Review:**

- The main strength of this paper lies in its conceptually simple but very effective idea of latent label encodings, which is quite different from all previous methods using actual label encodings. The new LLEM method is simple and straightforward, and it naturally eliminates some ad-hoc designed tasks in the previous state-of-the-art method LACO, but surprisingly achieves better performance.

- The latent label encoding reminds me of the P-tuning for acquiring a best-performing prompt (https://arxiv.org/pdf/2103.10385.pdf). Since neural networks are inherently continuous, discrete prompts can be sub-optimal. The P-tuning also leverages trainable continuous prompt embeddings to serve as prompts fed as the input to the pre-trained language models, and then optimizes the continuous prompts using gradient descent as an alternative to discrete prompt searching. They are two completely different NLP tasks, but seem to share some common insight, and they might provide a new line of thought for some interesting future research.

- Figure 2 shows the LLEM outperforms LACO in every single group of data (divided by label frequencies or numbers), but corresponding discussions just present the observations of experimental results, and the better performance does not seem to easily lead to the conclusion that LLEM is superior in label-correlation utilization. As mentioned by the authors, some further work are still required to "investigate how the latent-label correlate with each other, with the actual labels and the context, and how can they be interpreted".

- The paper was well written and organized, and there is an extensive survey on recent work of multi-label text classification.

- Minor issues
    - typos: "intresting" on Page 9


**Summary Of The Paper:**

This paper proposes a novel multi-label text classification method named LLEM that jointly encodes a document and latent labels (with smaller number than actual labels), and tries to better model label correlations implicitly and impose less a priori limits compared with previous state-of-the-art works. The method is conceptually simple but outperforms the state-of-the-art results on two widely used benchmarks.

**Summary Of The Review:**

- It is an interesting and useful study on multi-label text classification.

---

### Official Review · Reviewer_fS9V · 2021-11-02

**Correctness:** 3
**Technical Novelty And Significance:** 2
**Empirical Novelty And Significance:** 1
**Recommendation:** 3
**Confidence:** 4

**Main Review:**

## Strengths:
* The model has been tested and compared against the LACO algorithm [1] that sets the SOTA on two widely-used multi-label text classification datasets: AAPD and RCV1-V2, and outperforms LACO using Hamming Loss and Micro-F1.
* Especially, the proposed method has even better performance than the baseline LACO algorithm, on the low-frequency labels and intensive-label samples.

## Weaknesses:
* The paper's overall contribution and impact seem quite limited. The idea  is very similar to LACO [1], and the paper’s contribution is that it encodes the latent label rather than the actual label.
* Baseline models: There are many recent papers on XMC (extreme multi-label classification), investigating how to utilize the label correlations. Although these papers focus on the MLC cases where there are thousands or millions of labels. These methods are also applicable to general MLC task. ECLARE [2] and vanilla BERT-based AttentionXML [3], without the tree-based hierarchical structure, might be very strong baseline models. I am curious to know the performance of these models. I might have misunderstood something here. The author can clarify the choice of baseline models or clarify my misunderstanding in the rebuttal.
* Some questions on the algorithm itself:
   * Label semantics is not utilized in the model design since the embeddings of the latent labels are randomly initialized.
   * How to interpret the latent labels?
   * K, the number of latent labels, is a hyperparameter. But how to choose K? There is no ablation study on tuning k in the experiments section.
   * The authors mentioned LACO can be less effective and even impossible when the number of actual labels are very large. However, the authors only show the experimental results of AAPD and RCV1-V2 datasets, which have 54 and 103 labels respectively. There are popular datasets [4] in the XMC community, for example RCV1-2K, EURLex-4K and AmazonCat-13K. If the authors can show the proposed method is applicable to these XMC datasets as well. The advantage over LACO can be justified and the model can have even bigger impact in the XMC community.

1. Enhancing Label Correlation Feedback in Multi-Label Text Classification via Multi-Task Learning
2. ECLARE: Extreme Classification with Label Graph Correlations
3. AttentionXML: Label Tree-based Attention-Aware Deep Model for High-Performance Extreme Multi-Label Text Classification
4. http://manikvarma.org/downloads/XC/XMLRepository.html


**Summary Of The Paper:**

The paper presents a method that uses latent label representations to model label correlations implicitly, for the multi-label text classification (MLTC) task. The method concatenates a set of randomly generated latent labels to input text tokens. Then the method uses this as the input to the BERT model. At last, the contextual encodings of these latent labels are used to generate predictions for the actual labels.
The model has been tested and compared against the LACO algorithm [1] that sets the SOTA on the AAPD and RCV1-V2 datasets, and outperforms LACO using Hamming Loss and Micro-F1. Especially, the proposed method has even better performance than the baseline LACO algorithm, on the low-frequency labels and intensive-label samples.

1. Enhancing Label Correlation Feedback in Multi-Label Text Classification via Multi-Task Learning

**Summary Of The Review:**

I would like to reject this paper since overall contribution and impact seem quite limited. Also, the authors neglect recent advances in related fields.

---

### Official Review · Reviewer_BUhV · 2021-11-02

**Correctness:** 3
**Technical Novelty And Significance:** 2
**Empirical Novelty And Significance:** 2
**Recommendation:** 3
**Confidence:** 4

**Main Review:**

Recent interest for multi-label classification lies in how to model the label correlations. Instead of modeling the label correlations explicitly, this paper suggests that it is more effective to implicitly model the label correlations via latent labels.

The reported experimental results show that the proposed LLEM outperforms the other baselines. And the empirical study shows that pretraining latent label embeddings cannot improve the results. If I understand correctly, the latent labels can be considered as the expanded [CLS] tokens prepended to the sentences/tokenized text. Since the [CLS] token is intended to represent the sentence, pretraining the latent label embeddings may not capture the label correlations and lead to performance loss.

What concerns me most is how the classification benefits from the latent labels. Modeling the label correlations explicitly is intuitive and explainable, such as the label embedding methods. It would be interesting to have more empirical analysis, such as how the number of latent labels influence the classification results.

In addition, the paper mentions the label-correlation overload, but lack of further discussions. More detailed theoretical and empirical study could be given, such as how LLEM learns the label correlations in an adaptive manner with few a priori limits.

**Typo**

*Given a text T containing m sequential word tokens w_0, w_1, . . . , w_{m−1}, w_i*, the subscript of the last token should be *m*

**Summary Of The Paper:**

This paper addresses the task of multi-label text classification by modeling the label correlations implicitly. Different from the previous works that explicitly model the label correlations, such as the label embedding methods, this paper proposes modeling the label correlations via latent labels. The proposed method outperforms the baselines on two multi-label text classification benchmarks in the reported experimental results.

**Summary Of The Review:**

This papers proposes a simple method to model the label correlations implicitly via latent labels. The reported experimental results show the proposed method is effective. Whereas the method lacks of theoretical and empirical analysis. In summary, this paper is not good enough for ICLR.

---

### Official Review · Reviewer_FMut · 2021-11-02

**Correctness:** 2
**Technical Novelty And Significance:** 1
**Empirical Novelty And Significance:** 2
**Recommendation:** 3
**Confidence:** 4

**Main Review:**

- Technically, the idea of appending multiple "latent tokens" to the beginning of each document is not new. As far as I know, it was first proposed in [1] (the BERT-XML model) for multi-label text classification and later in [2]. Note that in both papers, this is just a small trick instead of their key novelty, so there may be even earlier work already proposing this trick. That being said, the novelty of this submission is rather limited.

- It is quite confusing why these "latent tokens" can be claimed as "labels" and can implicitly model label correlations. Essentially, they can be viewed as multiple [CLS] tokens or "probes" to text semantics. In [1] and [2], their explanation is that the label space is quite large (e.g., >10K), so one [CLS] token (i.e., a 768-dimension vector) may not be informative enough to predict the relevant labels. In your paper, I cannot see intuitive explanations or experiments supporting your claim of "implicit label correlations".

- Some important baselines are missing, including XML-CNN [3] and AttentionXML [4].

- Significance tests are missing. It is unclear whether your improvement in Tables 1 and 2 is statistically significant or not. Please run each experiment multiple times and report standard deviation or p-values.

- In Section 4.2, it is confusing to append true labels to the input text sequence during training. No doubt this cannot perform well because the encoder already knows the labels and will not focus on the remaining part. However, during inference, you no longer have such label information.

[1] Xun et al., Correlation Networks for Extreme Multi-label Text Classification. KDD 2020.

[2] Zhang et al., MATCH: Metadata-Aware Text Classification in A Large Hierarchy. WWW 2021.

[3] Liu et al., Deep learning for extreme multilabel text classification. SIGIR 2017.

[4] You et al., Attentionxml: Label tree-based attention-aware deep model for high-performance extreme multi-label text classification. NeurIPS 2019.

**Summary Of The Paper:**

This paper proposes to implicitly model label correlations in multi-label text classification. Different from previous studies (e.g., tree-based models) that describe label correlations explicitly, this paper appends "latent labels" to the beginning of each document and feeds it into a BERT classifier. These "latent labels" are randomly initialized, and the concatenation of their output is used for classification.

The authors conduct experiments on two benchmark datasets, AAPD and RCV1. Experimental results show that their proposed method outperforms several baselines for multi-label text classification. Ablation studies further show that using "latent labels" is better than using "actual labels" in their framework.

**Summary Of The Review:**

The key idea has been proposed in previous studies, so the novelty is rather limited. The intuition is not well explained.

---

### Official Review · Reviewer_82Sn · 2021-11-08

**Correctness:** 2
**Technical Novelty And Significance:** 1
**Empirical Novelty And Significance:** 2
**Recommendation:** 5
**Confidence:** 4

**Main Review:**

## Strengths:
- encouraging experiment results that show marginal gains over baselines on two small scale XMTC datasets

## Weakness
- lack of complexity analysys. What is the time complexity with respect to number of labels during training and inference?
- How scalable is the proposed method? Can it be used in extreme multi-label classificiation problems where number of labels is million or more? Such experiments are missing.
- The rationale of latent labels are not convincing. What does the latent labels mean? Does it serve as a semantic cluster that consists of multiple labels? How do you find such latent labels? Otherwise, how can you claim you are modeling the correlation between labels?
- The experiment section should also mention the difference in model parameters. Are the proposed methods using more number of parameters?



**Summary Of The Paper:**

This paper consider multi-label text classification problem and propose a cross attention Transformer encoder to model the correlation between latent labels and input text sequence. The hidden states of those latent labels are concatnate as input to layers of MLP for the classification head. The experiment results show marginal gain over the baselines.

**Summary Of The Review:**

This paper propose a cross-attention Transfomrer architecture for the multi-label text classification problem. Such modeling architecture is not new in MLTC literature. In addition, the rational of latent labels are not techincal sound and justified. Time complexity with respect to number of labels is also missing. Finally, the author should consider experiment on large-scale multi-label datasets such as Wiki-500K and Amazon-3M. Given those reasoning, I would vote this paper being marginal below the acceptance threshold.

---

### Decision · Program_Chairs · 2022-01-20

**Decision:**

Reject

**Comment:**

This paper proposes an approach for multi-label text classification. The method constitutes appending few "label" tokens to the beginning of the text input instead of the traditional single <CLS> token. The paper shows improvements over a competitive baseline on two datasets.

Reviewers agree that the novelty and contribution of the paper are marginal. The method of appending extra "fake" tokens has been used in other works as a "trick". It is also unclear how adding a few extra tokens allow for the model to represent label dependencies better.

The authors did not respond to the reviews, so there was no further dicussion.